# A Matter of Size and Placement: Varying the Patch Size of Anisotropic Patchy Colloids

**DOI:** 10.3390/ijms21228621

**Published:** 2020-11-16

**Authors:** Carina Karner, Felix Müller, Emanuela Bianchi

**Affiliations:** 1Institut für Theoretische Physik, TU Wien, Wiedner Hauptstraße 8-10, A-1040 Wien, Austria; felix.mueller@live.at; 2CNR-ISC, Uos Sapienza, Piazzale A. Moro 2, 00185 Roma, Italy

**Keywords:** two-dimensional assembly, crystalline monolayers, patchy platelets

## Abstract

Non-spherical colloids provided with well-defined bonding sites—often referred to as patches—are increasingly attracting the attention of materials scientists due to their ability to spontaneously assemble into tunable surface structures. The emergence of two-dimensional patterns with well-defined architectures is often controlled by the properties of the self-assembling building blocks, which can be either colloidal particles at the nano- and micro-scale or even molecules and macromolecules. In particular, the interplay between the particle shape and the patch topology gives rise to a plethora of tilings, from close-packed to porous monolayers with pores of tunable shapes and sizes. The control over the resulting surface structures is provided by the directionality of the bonding mechanism, which mostly relies on the selective nature of the patches. In the present contribution, we investigate the effect of the patch size on the assembly of a class of anisotropic patchy colloids—namely, rhombic platelets with four identical patches placed in different arrangements along the particle edges. Larger patches are expected to enhance the bond flexibility, while simultaneously reducing the bond selectivity as the single bond per patch condition—which would guarantee a straightforward mapping between local bonding arrangements and long-range pattern formation—is not always enforced. We find that the non-trivial interplay between the patch size and the patch position can either promote a parallel particle arrangement with respect to a non-parallel bonding scenario or give rise to a variety a bonded patterns, which destroy the order of the tilings. We rationalize the occurrence of these two different regimes in terms of single versus multiple bonds between pairs of particles and/or patches.

## 1. Introduction

Two-dimensional materials are becoming increasingly important owing to the variety of technological and biomedical applications for which they can be used [1,2,3,4,5]. According to the size of the building units, ranging from molecules and macromolecules to nano- and micro-particles, surface patterns can be designed to achieve specific functionalities at different length scales. The formation of well-designed surface structures can often be described by a combination of general ingredients. For instance, the assembly of organic molecules functionalized with multiple carboxyl groups on a highly oriented pyrolytic graphite surface [6,7,8,9,10,11] can be rationalized in terms of the rotational symmetry of the building units, their aspect ratio, the placement of the bonding groups, and possible energy differences between binding groups [12,13,14,15]. These same general ingredients were proven to be crucial in other molecular-scale processes: by combining the anisotropy of the building blocks with a low bond functionality, it is possible, for instance, to control the emergence of crystal polymorphs in protein systems [16] or to separate the enantiomers of chiral molecules [17]. The same approach—non-spherical building units provided with a discrete number of bonding sites—can be also applied at the nanometer scale: DNA origami of different shapes have been programmed to assemble into prescribed two-dimensional tilings by taking advantage of blunt end stacking and hybridization sites, thus providing versatile platforms to engineer optical metamaterials and biomimetic tissues [18,19,20,21,22]. At even larger length scales, micrometer non-spherical colloids decorated with attractive spots along their perimeter also form two-dimensional aggregates whose complex geometries can be related to the properties of the constituent units [23,24,25]: close-packed versus porous, surface structures, as well as finite clusters with specific architectures can be designed by tailoring the single particle features [26,27,28,29,30]. Colloidal platelets with non-spherical shapes and directional bonding sites—often referred to as patches—constitute an ideal playground for testing and understanding the driving mechanisms of two-dimensional tilings. The self-assembly of patchy colloidal platelets is governed by a competition between shape and bonding anisotropy: on the one hand, non-spherical hard shapes tend to maximize edge-to-edge contacts [23,31,32]; on the other hand, the presence of attractive patches imposes additional constraints on the assembly process, thus leading to a rich aggregation scenario [24,25,26].

We recently considered systems of anisotropic patchy platelets with a small number attractive patches placed in different arrangements along the particle edges, and we rationalized the emerging aggregation behavior in terms of the complex interplay between steric incompatibilities and (un)satisfied bonding geometries [27,28,29]; additionally, by tuning the patch placement on the particle edges, the combination of shape and bond anisotropy was used to drive the assembly from a close-packed arrangement to tunable porous lattices with the same geometric pattern [27]. We now aim at investigating the effect of the patch size, i.e., of the attractive interaction range of the bonding sites, on the two-dimensional tiling of the same anisotropic patchy colloids.

Within the vast realm of currently available patchy colloids and platelets [23,33,34], the patch size can be tuned in a wide variety of systems by changing the physical or chemical features that characterize the patches, such as the patch surface extension [35,36,37], the patch surface roughness [38], the length of the DNA strands and ligands grafted onto the particle surface [24,39], as well as the pH or the salt concentration of the colloidal dispersion [40,41]. The effect of the patch size for spherical particles and colloidal molecules has been investigated in the assembly of finite clusters of various symmetries [42,43,44] and in the spontaneous nucleation of two- and three-dimensional crystals, such as the diamond lattice [45,46] and the honeycomb network [47]. In general, the assembly mechanism of these systems is governed by a compromise between selectivity, kinetic accessibility, and thermodynamics. On the one hand, small patches impart a strict directionality to the inter-particle bonds, thus selecting the target structure that is compatible with the fully bonded local arrangement of the building units; on the other hand, the narrower the patches are, the more difficult it is for the particles to rearrange an incorrect aggregate, and the system is likely kinetically trapped. In contrast, large patches introduce a certain degree of flexibility in the bonding mechanism: relaxed bonding restrictions favor local rearrangements—and the formation of the equilibrium structure—but at the same time, bonds become increasingly less selective with the patch size, thus leading to disordered assemblies. It is worth noting that, when patches can form at most one bond, there exists a correspondence between the patch number and the bonding pattern that provides the control over the features of the target crystal or finite cluster. Nonetheless, even when the single bond per patch condition is satisfied, an extreme flexibility of the bonding mechanism can still disfavor the stabilization of ordered phases at any value of the interaction energy [48].

In the present contribution, we consider a system of regular hard patchy rhombi (of edge length *l* and opening angle α=60°) decorated with four identical patches (of size *r* placed in different arrangements along the particle perimeter (see Figure 1). The patch arrangements are determined by a predefined topology and a parameter Δ, which sets the exact patch positions along the rhombi edges (see again Figure 1): pairs of patches on adjacent edges are at distance Δ from their closest vertex, while pairs of patches on opposite edges are described by (Δ,1−Δ); when Δ≠0.5, the chosen topology is referred to as asymmetric (note that it was labeled feq-as2 in Reference [27]), while when Δ=0.5, each patch sits in the center of its respective edge, and the resulting symmetric arrangement is referred to as the center topology. We note that the chosen patch topology is symmetric with respect to Δ=0.5 when mirrored along the major axis of the rhombi. Due to their steric incompatibilities, rhombic platelets align with each other in two possible orientations: parallel (p) and non-parallel (np); depending on the patch placement defined by Δ, these alignments can be realized by edge-to-edge (on-edge) contacts or in staggered (off-edge) configurations (all represented in Figure 1). If two of these neighboring particles are additionally bonded together via at least one of their patches, they are considered to be members of the same cluster. When all patches are in the center of their respective edge, edge-to-edge—either p or np—configurations are energetically favored, thus leading to a close-packed random (r) tiling: the r-tiling emerges because our patchy rhombi are equally likely to bond in a p- or np-fashion; nonetheless, it is worth noting that, as there are bonding restrictions on particles attaching to already formed clusters, the total number of p- and np-bonds in the resulting r-tiling is not equal [27]. In contrast, on decreasing Δ, staggered configurations—again, either p or np—are energetically preferred, thus leading to the formation of pores in the assembling tilings. The emerging porous tilings are characterized by either rhombic pores (p-bonded domains) or by hexagonal and triangular pores (np-bonded domains) [27]. Additionally, for the chosen patch topology, porous p- and np-clusters grow next to each other within the same sample, but as p- and np-domains are more and more incompatible on decreasing Δ, a switch to an np-tiling eventually occurs at extreme Δ-values [27]. The radial distribution functions that characterize np-, p-, and r-tilings are denoted as g(R) with *R* as the center-to-center particle distance and are reported in Figure 2, together with their corresponding simulation snapshots. We note that, for the asymmetric patch topology with Δ=0.2, the first-neighbor peak of the np-pattern is at R/l>1, while the corresponding peak of the p-pattern is at R/l<1; in contrast, for the symmetric patch topology, the p-peak is at R/l>1 and the np-peak at R/l<1. Additionally, we note that although the co-occurrence of p- and np-bonding motifs does not create a lattice order, the g(R) of the r-tiling has sharp peaks as typical distances do repeat.

While in Reference [27], the patch size was set to a few percent of the particle edge length [12], in the present contribution, we investigate the effect of larger and larger patches on the tiling process. As large patches are less selective, the restrictions induced by the bonding anisotropy are expected to become more relaxed, meaning that the ordered—mostly np—tiling scenario observed at small *r*-values might be destroyed on increasing the patch size. Nonetheless, shape anisotropy still plays a role in the assembly process, thus possibly leading to a complex interplay between flexible bonds and non-spherical particle shapes. Such an interplay has to be rationalized in view of emerging poly-bonds: patches, as well as particles are expected to bond in multiple ways as soon as the patches allow for a high flexibility in both bond movements and bonding restrictions.

## 2. Results

Our patchy rhombi model—first introduced in Reference [27]—is detailed in Section 4 with the particle parameters reported therein. To simulate the two-dimensional assembly process of these building units on a non-interacting substrate, we perform grand canonical Monte Carlo simulations; the details of the simulations are reported in Section 4 together with the simulation parameters.

We begin our discussion with a qualitative, visual analysis of the typical morphologies observed in the emerging tilings on increasing the patch size *r* at different Δ-values. Figure 3 provides an overview of simulation snapshots at selected (Δ,r)-values. The multiple colors of the particles, indicating their orientations, help visualize differently ordered domains. At r=0.05, we obtain the same tilings as observed in Reference [27], i.e., a porous np-tiling for the most asymmetric topology (see Figure 3a,m) and an r-tiling for the center topology (see Figure 3i,p). In between these two regimes, we observe a coexistence of p- and np-domains (see Figure 3e,n). We note that, on reducing the asymmetry of the topology, the hexagonal/triangular, as well as rhombic pores shrink, leading from open to close-packed tilings. Moreover, the more the patches move towards the center topology, the more commensurable the p- and the np-tiling become, and the higher the probability to find p- and np-domains within the same system. Interestingly, at r=0.08, we observe coexisting p- and np-domains already at Δ=0.2 (see Figure 3b), as an effect of the larger patch radius that enhances commensurability and promotes p-bonding—as detailed in the following. On reducing the asymmetry of the topology, p- and np-domains can still be identified: at Δ=0.425, they form a mixed bonding motif (see Figure 3f,q) that develops into an r-tiling at Δ=0.5 (see Figure 3j). A different picture presents itself as the radius is further increased to r=0.15 and then to r=0.2. In many of these domains, two new phenomena appear (see the discussion of Figure 4): (1) pairs of particles can be bonded to each other via more than one patch (they are thus referred to as pairs of poly-bonded particles), and (2) a patch can form a bond with more than one patch (it is thus referred to as a poly-bonded patch); in the discussion of Figure 5, we describe the impact of these bonding scenarios on the tilings in detail. In general, the poly-bond scenarios seem to favor a p-alignment between the particles (see Figure 3o,r,s), as well as the emergence of mixed bond types: as more orientations are possible, more bonded motifs are allowed (see Figure 3c,d,g,h,l).

Even though both poly-bond scenarios have an impact on the tiling process, we first focus on the effects related to the emergence of bonding configurations where a single patch can bond to two or more patches at the same time. Due to the anisotropy of our building blocks, the interplay between the patch size and the patch position is highly non-trivial: the whole (Δ,r) parameter space must be spanned to detect where the single bond per patch condition is no longer guaranteed. Figure 4a displays a heat map of the maximum number of bonds per patch over the whole range of (Δ,r)-values. Clearly, bigger patches favor the formation of many bonds per patch, but we find that the first occurrence of poly-bonds per patch is strongly dependent also on Δ. While at Δ=0.2, the first poly-bonds per patch occur already at r=0.08, at Δ=0.5, the first poly-bonds per patch appear only at r=0.17. This can be rationalized by taking into account the effect of the particle shape: for extremely asymmetric patch topologies, the patches are closer to the corners of the rhombi, thus making themselves accessible to more than one patch, if the patch size is large enough; in contrast, the closer the patch position is to the center, the more the rhombi edges shield the patch and the larger *r* needs to be to allow for poly-bonds. We observe that the critical patch size for which the first poly-bonds start to appear scales with $∼\frac{\Delta }{2}$. We also observe that, as soon as the geometric factors allow for the formation of multiple bonds per patch, the fraction of patches that form poly-bonds in the system grows sharply for any further increase of *r*. Figure 4b shows the average fraction of patches that formed poly-bonds as a heat map in the (Δ,r)-plane, while Figure 4c shows the average poly-bond per patch ratio as a function of *r* at selected Δ-values. For Δ=0.2, the average fraction of poly-bonded patches goes up to 0.9 at the maximum *r*-value considered here, while at the same *r*, the fraction of poly-bonds per patch at Δ=0.45 is about 0.5. It is crucial to note that the maximum number of patches that are allowed to bond at the same site has a drastic effect on the particle arrangement, as shown in Figure 4d, where snapshots of typical patch configurations at Δ=0.2 and different *r*-values are displayed. When patches can bond with up to two/three other patches, a p-bond scenario is promoted (see the first and the second panel of Figure 4d), while patches are able to bond with four/five patches only in five- and six-star configurations, which are np-bond motifs (see the third and the fourth panel of Figure 4d). The bonding motifs favored by the described poly-bonds per patch impact the ordering of the self-assembled tilings heavily, as described in the following.

We quantify the ordering of a tiling via a suitably designed order parameter Ψ (see Section 4) that takes values between −1 (indicating that all neighbors of all particles are oriented in an np-fashion) and 1 (indicating that all neighbors of all particles are oriented in a p-fashion); a perfect random tiling corresponds to Ψ=0. We note that Ψ can fluctuate around zero for different reasons: if the system is in a fluid phase, if the system forms a perfect random tiling, or if domains with different bonding types develop during the assembly process. To differentiate between these cases and better characterize the spatial arrangements of the different tilings, we also calculate the radial distribution function, g(R), where *R* is the center-to-center distance between two rhombi.

In Figure 5a, we report the average Ψ as a function of Δ at different *r*-values. For r=0.05, we confirm—as suggested by the snapshot analysis and previous results [27]—that the more asymmetric the topology is, the more the np-tiling is favored: for Δ≤0.4, Ψ<−0.75. It is worth noting that, for Δ=0.25,0.3, and 0.35, parallel simulation runs can either form a porous np-tiling with Ψ<−0.9 or a porous p-tiling with Ψ>0.9: the latter appears less often than the former, as highlighted in Figure 5b, where histograms for Ψ are reported (see the green histogram corresponding to r=0.05 and Δ=0.25); this behavior results in a bimodal distribution and explains the large error bars on Ψ for these systems. When Δ is increased beyond Δ=0.4, Ψ grows monotonically towards zero, because p- and np-tilings become more commensurable (see the red histogram corresponding to r=0.05 and Δ=0.45 together with Figure 3e) until the random tiling emerges (see the red histogram corresponding to r=0.05 and Δ=0.5 together with Figure 3i). On increasing the patch size beyond our reference value r=0.05, the enhanced angular freedom in bonding leads to the formation of more mixed configurations either as coexisting p- and np-clusters (see Figure 3) or as mixed bond types (see Figure 4d). At r=0.08, p-domains are promoted with respect to np-ones, as highlighted by the shift of Ψ towards zero at all Δ-values (see Figure 3b and the corresponding green histogram in Figure 5b). This behavior can be associated with the higher commensurability of the porous p- and np-tilings—due to the larger patches—promoting the parallel growth of these two types of domains in the same sample (see Figure 3b). On further increasing *r*, the interplay between the patch size and the patch position becomes quite complex: the specific trends of Ψ as a function of Δ at selected *r*-values (see Figure 5a) can be related to the presence of different poly-bonds per patch in the system. At r=0.10, p-domains are largely promoted for the most asymmetric patch topology. This phenomenon can be understood by taking into account the occurrence of poly-bonds per patch (see Figure 4b): at Δ=0.2, a fraction of 0.4 of all patches bond with up to three patches, while at larger Δ-values, poly-bonds are still negligible, and Ψ approaches zero quickly. The visual analysis of particle arrangements when a patch can bond to two other patches (see the first and second panels of Figure 4d) suggests that this bonding scenario promotes the p-alignment between the rhombi. The same bonding scenario can be observed at r=0.12: p-domains are largely promoted up to Δ=0.25, where a fraction of 0.3 of all patches bond with two or three patches (see again Figure 4a,b), while Ψ≈0 at larger Δ-values where poly-bonds per patch are not present in a significant amount. In general, when r>0.10, Ψ develops a maximum, and the corresponding Δ-value increases on increasing *r*: for the largest patches, i.e., for *r* = 0.2, Ψ grows monotonically from zero at Δ = 0.2 to almost Ψ≈0.5 at Δ = 0.45, signaling a tendency to form p-domains (see the red histogram corresponding to r=0.2 and Δ=0.45 in Figure 5b); as soon as Δ>0.45, Ψ drops again towards zero. This peculiar behavior stems from the occurrence of different poly-bonds per patch in the system: when the poly-bonds per patch are up to two/three, p-bonds are promoted (first and second panels of Figure 4d), while when the poly-bonds per patch are up to four/five, np-bonds are favored again within the five-star and six-star motifs (third and fourth panels of Figure 4d). When the patch topology is such that the single bond per particle condition is recovered at the given patch size *r*, Ψ tends towards zero.

As anticipated above, to better characterize the change in the ordering of the tilings on increasing the patch size, we also consider the radial distribution function. While the ordered tilings are expected to show significant peaks and troughs, non-crystalline structures are expected to show a less pronounced behavior and quickly converge to one. Figure 6a shows the g(R) for the most asymmetric patch topology at selected *r*-values. We observe that, for r=0.05, g(R) has the sharp peaks of an ordered hexagonal tiling, corresponding to an np-pattern (see Figure 2 as a reference). Already at r=0.08, the peaks are significantly less pronounced; at short inter-particle distances, we observe the appearance of the peak related to p-bonded particles (see again Figure 2 as a reference), while at large R/l-values, the peaks vanish, indicating a lower long range order. At r=0.15, all peaks but those indicating the nearest neighbors have vanished, indicating a complete loss of long range order. We note that the p-peak at small inter-particle distances is in this case more pronounced than the np-peak, confirming the predominance of p- over np-bonds at this (Δ,r)-combination. For the central topology—reported in Figure 6b—we observe sharp peaks for r=0.05, which correspond to an r-tiling (see Figure 2 as a reference). Again, for r=0.08, g(R) has less pronounced peaks, and at r=0.15, the only well-pronounced peak of the g(R) is the one indicating the first neighbor shell; we note that this peak is relatively broad, as the many different bonding motifs (see Figure 4d) imply the existence of many possible distances for the first neighbors. In summary, for both patch topologies, large patch radii, i.e., high bond flexibilities, destroy the order of the tilings, as at the largest bond flexibility, edge-to-edge alignments are no longer crucial for the bond formation.

We quantify the change in bond flexibility with increasing *r* at different values of Δ with the bonding volume $V_{b}$, i.e., the volume in configuration space occupied by all possible relative positions and orientations a particle can assume while staying bonded to its neighbor. Figure 7a,d shows $V_{b}$ at different *r*-values as a function of Δ for p- and np-bonds, respectively. Sketches of the dimers used in the calculations (see Section 4) are reported below the corresponding panels; note that we numerically evaluate $V_{b}$ for Δ∈[0.2,0.8], while in the self-assembly simulations, we consider Δ∈[0.2,0.5]. This is due to the fact that, while the asymmetric and center topologies for four patches are mirrored with respect to Δ=0.5, the parallel asymmetric topology for one patch is not: for Δ<0.5, bonded patches are closer to the small angles; for Δ>0.5, bonded patches are closer to the big angles. Of course, in the four-patch systems, both bonding types are observed. We find that, for both p- and np-bonds, $V_{b}$ is minimal at Δ=0.5 and increases monotonically towards Δ=0.2 and Δ=0.8. This can be understood intuitively by considering that the more off-center the patches are placed, the more the bonded particles are allowed to wiggle. On increasing *r*, $V_{b}$ grows at all Δ-values, with the bonding volume remaining the highest at Δ=0.2 and Δ=0.8 (corresponding to the most asymmetric patch topologies) and the lowest at Δ=0.5 (central patch topology). To better characterize the growth of $V_{b}$ with *r* at all Δ-values, we represent $V_{b}$ in Figure 7b,e as a function of *r* at different Δ-values for the p- and np-dimers, respectively. For both p- and np-bonds, the increase in $V_{b}$ is the steepest at intermediate *r*-values, roughly between 0.08 and 0.14, while it is flatter at small (r<0.08) and large (r>0.14) *r*-values. Both the growth trend and the absolute growth in $V_{b}$ are similar for p- and np-bonds. In contrast, $V_{b}$ is asymmetric with respect to Δ=0.5 for p-bonds, whereas it is symmetric for np-bonds. These conclusions can be already drawn from Figure 7a,d and are quantified in Figure 7c,f, where we display $\left[V_{b}\left(\Delta \right)-V_{b}\left(1-\Delta \right)\right]/V_{b}\left(\Delta \right)$ at selected Δ-values. We observe that for p-bonds, the asymmetry grows with *r* to the point that, at Δ=0.2, $V_{b}$ is approximately 7.5% larger than $V_{b}$ at Δ=0.8 for large *r*-values, while for np-bonds, we do not observe a significant asymmetry (see Figure 7f). We note that, as the effective interaction energy of a single patch is proportional to $V_{b}$, a higher bonding volume implies a higher effective energy, favoring the parallel growth of multiple domains and the formation of grain boundaries. Furthermore, a higher bonding volume contributes to the emergence of poly-bonds especially at extreme Δ-values. The reason for this is geometrical: off-center patch positions are more accessible, and as soon as—for the given patch topology—the bonding volume is large enough, more than one patch can bond at the same site.

At this point, it is important to describe the effect of the emerging poly-bonds per particle pair as they impact both the bond flexibility and the ordering of the tilings. Figure 8a shows for which (Δ,r)-values p- (turquoise) and np- (burgundy) poly-bonds can occur, where p and np refers to the initial bonding state of the particle pair at the start of a Monte Carlo simulation (see Section 4). It is important to stress that—at a given (Δ,r)-combination—poly-bonded particle pairs can be both p- and np-bonded, and by virtue of this, the orientational alignment of the particles can change during the simulation with respect to its starting alignment (see Figure 8b–k). As p- and np-configurations can interchange during the simulation, we refer to the single-bond per particle pair scenario as the one resulting from the overlap between the two shaded areas in Figure 8a. The figure suggests that poly-bonds per particle pair form because of purely geometric factors: patches in a more off-center position favor the formation of poly-bonds per particle pair at smaller *r*-values with respect to patches in more centered positions. For the studied assemblies, poly-bonds per particle pair often co-occur with poly-bonds per patch: this is visualized by the snapshots labeled with the triangle and the trapezoid in Figure 4d, where poly-bonds per particle pair of the type depicted in Figure 8i co-occur with poly-bonds per patch. Thus, in the Δ-dependent regime of *r*-values where the single-bond per particle pair condition is no longer guaranteed, the formation of multiple bonds between pairs of particles promotes the emergence of p-domains. Nonetheless, it should be noted that, at any value of Δ, poly-bonds per particle pair only appear at a value of *r* larger than the *r*-value corresponding to the sharp rise in p-bonds, suggesting that the previously observed predominance of p-patterns on increasing the patch size is initiated by the poly-bonds per patch.

As anticipated, poly-bonds per particle pair have a very strong effect on the bond flexibility: when two particles bond together twice—and this happens on increasing *r*—the growing bond flexibility is reduced by the double bond between the particles. The reduction of the bond flexibility as soon as poly-bonds per particle pair occur is shown by the histograms of the relative particle orientations, P(ΔΩ), defined in Section 4. For Δ=0.2 (see Figure 9a), P(ΔΩ) for r=0.05 shows the typical narrow peaks at π/3 and 2π/3 of an exclusively np-tiling; as *r* increases to 0.08, the np-peaks become wider, and the peaks corresponding to the emerging p-bonds appear at π and zero. At the same time, all peaks become broader as a consequence of the higher bonding flexibility. As *r* increases further into the poly-bond per particle pair regime, i.e., at r=0.11, we observe a narrowing of the peaks, which indicates a loss of orientational flexibility for a large proportion of bonds in the systems. Note that now the peaks at π and zero are more pronounced than those at π/3 and 2π/3 as there are more p-bonds. The most typical configuration is shown in Figure 8i. With a further increase of *r* to 0.2, the peaks widen again, but remain more pronounced than those at *r*-values corresponding to the single-bond per particle pair regime (at r=0.08). For Δ=0.45 (see Figure 9b), we observe the same trend, with the narrowing of the peaks occurring at r=0.2, which is the patch radius at which the poly-bonds per particle pair first occur at this Δ-value. For Δ=0.5 and r=0.05, we observe four sharp peaks, indicating the random tiling phase. As *r* increases, these peaks become broader. At r=0.15, the peaks are still visible despite the high orientational flexibility of the bonds. The near absence of poly-bonds per particle pair (see Figure 8a) allows for orientations outside the peak positions.

## 3. Discussion

We consider a coarse-grained model where the shape and bond anisotropies are combined together by means of attractive patches displaced along the edges of undeformable particles and perform two-dimensional Monte Carlo simulations of their self-assembly process. This model—featuring rhombic platelets with four identical patches in different arrangements along the particle perimeter—was introduced to describe the tiling of tetracarboxylic acids on a highly oriented pyrolytic graphite surface [9,12,27]; nonetheless—owing to its fundamental character—the same model is also able to describe the behavior of anisotropic units at lager length scales, such as DNA origami nanoparticles with ligands grafted at specific sites along the particle edges or micrometer non-spherical colloids functionalized with attractive surface areas. Our results may thus provide insights into the tiling behavior of these three types of building blocks, namely tetracarboxylic acids, DNA origami with dangling ligands, and non-spherical patchy platelets. The patch size of our simple model can be then associated with, respectively, the characteristic length of the hydrogen bond in different molecular configurations, the length of the ligands on the DNA nanoparticles, and the patch interaction range of patchy colloidal platelets. As the hard steric repulsion plays a key role in the assembly mechanisms of our patchy rhombi, the reference model—with relatively small patch sizes—is more suitable to describe the second and the third classes of building units; these units can be engineered ad hoc: the single versus multiple bonding scenarios can be enforced at the particle synthesis level, thus guaranteeing a powerful tool to stir the assembly towards target tilings. In contrast, for the first class of building units, the steric incompatibilities of the model are needed to reproduce the directionality of the bonds between the carboxyl groups: rhombic units with four equal patches placed in the center of each edge were proven to provide a very accurate description of the assembly of TPTC molecules (p-terphenyl-3,5,3’,5’-tetracarboxylic acid) on highly oriented pyrolytic graphite [12]. As long as the single bond per patch condition is guaranteed, a larger patch size in the model releases the hard constraints, thus reducing the mismatch between hard shapes and molecular voids. In the literature, several experimental results are reported on the assembly of tectons of tetracarboxylic acids at the solid-liquid interface between highly oriented pyrolytic graphite and nonanoic/octanoic/heptanoic acids [6,7,8,9]. To the best of our knowledge, a unifying picture is still missing and several mechanisms are proposed to understand the rationale behind the emerging tilings, such as possible energy differences of the hydrogen bonds in different molecular configurations [6,8], solvent-induced polymorphism [49,50], or the possible effects of the dynamics of the molecules in solution. The release of the hard constraints—together with the placement of the bonding sites along the particle edges—may be used for a broader investigation of the tilings emerging in different systems of tetracarboxylic acids on the basis of purely geometric and energetic arguments.

## 4. Materials and Methods

### 4.1. Particle Model

We consider regular hard rhombi decorated with four identical attractive patches (see Figure 1). The aspect ratio of the rhombi—defined by edge length l=1 and the acute angle $\alpha =60^{\circ }$—is kept constant. Patches are arranged in a specific asymmetric topology (referred to as feq-as2 in Reference [27]) with respect to the perfectly symmetric arrangement—where each patch is located in the center of its respective edge. To switch from this center topology to an asymmetric topology, patches are gradually moved farther away from or closer to an arbitrarily chosen vertex by a quantity specified by the parameter Δ, which defines the patch relative position on its edge. In the chosen asymmetric topology, when one patch is at distance Δ from its reference vertex, its opposite patch is at distance (1−Δ) from the same vertex. Clearly, when Δ=0.5, the asymmetric topology corresponds to the symmetric (center) topology (see Figure 1). As the chosen patch topology is symmetric with respect to Δ=0.5 when mirrored along the major axis of the rhombi, it is enough to consider Δ≤0.5 (see Table 1 for a summary of the particle parameters).

The particle-particle interaction is a hard repulsion given by:$$U\left({\vec{R}}_{ij},\Omega_{i},\Omega_{j}\right)=\left\{\begin{array}{cc} 0 & \;\mathrm{if}\;i\;\mathrm{and}\;j\;\mathrm{do}\;\mathrm{not}\;\mathrm{overlap}\\ \infty & \;\mathrm{if}\;i\;\mathrm{and}\;j\;\mathrm{do}\;\mathrm{overlap} \end{array}\right.$$
where ${\vec{R}}_{ij}$ is the vector joining the centers of mass of particle *i* and *j*, while $\Omega_{i(j)}$ is the orientation of particle i(j). Overlaps between rhombi are detected via the separating axis theorem for convex polygons [51,52].

The patch-patch interaction is a square-well attraction given by:$$W\left(P_{ij}\right)=\left\{\begin{array}{cc} -\epsilon & \;\mathrm{if}\;P_{ij}<2r\\ 0 & \;\mathrm{if}\;P_{ij}\geq 2r, \end{array}\right.$$
where $P_{ij}$ is the patch-patch distance, 2r is the patch diameter, and ϵ corresponds to the patch interaction strength.

### 4.2. Simulation Details

To simulate the two-dimensional assembly of our patchy rhombi, we perform grand canonical Monte Carlo simulations with standard periodic boundary conditions. Together with single particle rotation/translation moves and particle insertions/deletions, we also allow for the collective movement of clusters by implementing cluster moves [53,54]. We equilibrate the systems for $10^{5}$ Monte Carlo sweeps at T=0.1 and a low packing fraction (i.e., at ϕ≈0.02) with $\mu^{eq}$, then we increase the chemical potential to $\mu^{*}$ to observe the assembly (see Table 2 for a summary of the simulation parameters). We run the simulations for about $10^{7}$ Monte Carlo sweeps before collecting statistics. In order to further improve statistics, we perform 16 parallel simulation runs per system. We start collecting statistics after the packing fraction has reached its plateau; depending on the system, this can take between $3\times 10^{6}$ and $17\times 10^{6}$ Monte Carlo sweeps.

We note that, in order to optimize the assembly of perfect surface structures, we choose r=0.05 as the reference point [27] and perform a systematic search for a chemical potential that promotes the growth of the most defectless tilings within a reasonable time period: selecting low values of μ can drastically increase the necessary simulation time or even make it impossible for any structures to develop, while, if the chosen μ is too large, the number of defects in the resulting system is likely to be high because the growth happens too fast. We select μ=0.14,0.15,0.175 and 0.2 (note that μ=0.25 in Reference [27]), and we systematically run simulations at different Δ-values. We quantify the number of defects—namely, the number of parallel defects in a non-parallel tiling—by plotting the average Ψ at different μ- and Δ-values (see Figure 10). Note that, as we want to investigate the effect of parallel defects within non-parallel tilings, we exclude configurations with mostly parallel tilings—i.e., with Ψ>0.8—that can occur at Δ=0.2−0.35 (see Figure 5 and the associated discussion). We find that at Δ=0.2, Ψ is lowest for μ=0.14 and μ=0.15, which indicates that these μ-values correspond to tilings with the most non-parallel bonds and the least parallel defects. Additionally, we note that for Δ=0.2−0.35, μ=0.175 and μ=0.2 exhibit a significantly larger standard deviation of Ψ compared to μ=0.14 and μ=0.15, which accounts for the fact that the configurations are highly variable—a further indication of the presence of large defects in some configurations. The visual inspection of the simulation snapshots also confirms that, at any Δ-values, the emerging tilings have the least amount of defects when μ=0.14; thus, we choose μ=0.14 to investigate the role of the patch size on the two-dimensional assembly. We note that, for the chosen μ-value, the fluid phase is always observed for r<0.05 at all Δ-values.

### 4.3. Order Parameter

We use the order parameter Ψ, first introduced in Reference [8] and designed to describe the randomness of the assembled tilings. It is defined as:(1)$$\Psi =\frac{0.608n_{p}-0.392n_{\mathrm{np}}}{0.608n_{p}+0.392n_{\mathrm{np}}}$$
where $n_{p}$/$n_{\mathrm{np}}$ is the number parallel/non-parallel bonded particles in the system, and the occurring numerical factors—estimated in numerical simulations [9]—are such that Ψ=0 for the perfect random tiling, while Ψ=1(-1) for a perfect parallel (non-parallel) one [27]. To determine if a particle pair is bonded in a parallel or non-parallel fashion, we calculate the relative orientation between the two particles, ΔΩ. In general, the relative orientation between two particles *i* and *j* is given by $\Delta \Omega =∥\Omega_{i}-\Omega_{j}∥$ and it is calculated for bonded neighbors. If 45°<ΔΩ<135°, we define the bond as non-parallel, otherwise if ΔΩ<45° or ΔΩ>135°, we define the bond as parallel.

### 4.4. Bonding Volume

We define the bonding volume, $V_{b}$, as the volume a particle occupies in the configuration space, while staying bonded to another particle. $V_{b}$ is calculated using a Monte Carlo simulation of two particles with a single patch initialized in a bonded state, either parallel or non-parallel, for Δ∈[0.2,0.8] and r∈[0.05,0.2]. Bonds are discouraged from breaking by setting $\epsilon =1000k_{B}T$. For all pair configurations in the bonded state, we calculate the relative patch-to-patch center distance vector $P_{ij,x},P_{ij,y}$ and the relative orientation of the particle pair, ΔΩ. This spans a three-dimensional configuration space $P_{ij,x},P_{ij,y},\Delta \Omega$. All observed configurations are binned into cubes of side length $l_{\mathrm{cube}}=0.001$. The bonding volume is then calculated by counting the number of occupied bins $N_{bins}$ and multiplying it by the volume of one cube $v_{\mathrm{cube}}=0.001^{3}$.

### 4.5. Definition of Poly-Bonds per Patch

The number of bonds per patch is defined as the number of different patches found within the bonding area of a patch, i.e., at a patch-to-patch distance $P_{ij}<2r$. In a single bond per patch scenario, there is only one other patch within the bonding area. Bonding configurations where a patch is bonded to two or more patches are called poly-bonded.

### 4.6. Definition of Poly-Bonds per Particle Pair

If two particles bond to each other via more than one patch, we define them to be poly-bonded as a particle pair. To calculate for which (Δ,r)-values poly-bonds per particle pair occur, we conduct a Monte Carlo simulation of two particles initialized in a single bonded state, either parallel or non-parallel, for Δ∈[0.2,0.8] and r∈[0.05,0.2]. The bond does not break because the interaction strength is set to $\epsilon =1000k_{B}T$. A pair of (Δ,r)-values corresponds to the poly-bond regime if after $10^{7}$ Monte Carlo sweeps, the particles have formed at least one additional bond.

### 4.7. Relative Orientational Distribution

The relative orientational distribution P(ΔΩ) is obtained for the particle orientations Ω∈[0,2π], where Ω=0 corresponds to the arbitrarily chosen reference orientation sketched in the corners of Figure 3. Note that to calculate P(ΔΩ), we consider the relative orientations between all bonded neighbors in a system. To achieve better statistics, we use 10 configurations each from all 16 parallel simulations runs for all (Δ,r)-values.

## 5. Conclusions

We consider the self-assembly of rhombic building blocks decorated with four identical bonding sites and present a systematic investigation on how an increasing flexibility in the bonding mechanism impacts the features of the surface structures formed by the selected units. We recently rationalized the two-dimensional aggregation of this class of systems by showing how the interplay between the particle shape and the patch topology gives rise to a rich variety of tilings, from close-packed to porous monolayers [27]. We here select a specific patch topology and study the effect of the patch size, i.e., of the attractive interaction range of the bonding sites, on the two-dimensional tiling mechanisms.

Due to their steric incompatibilities, rhombic platelets can align either in a parallel or in a non-parallel mutual orientation. For small patches placed at the center of their respective edge, a close-packed tiling emerges where particles arrange edge-to-edge in a random surface structure. As soon as these small patches are moved asymmetrically towards the rhombi vertices, porous domains where particles align either in a parallel or in a non-parallel pattern start growing next to each other until, for highly asymmetric patch topologies, the non-parallel tiling prevails. On increasing the patch size, bonds become more flexible and less selective at the same time: while in systems of spherical patchy particles, large patches mostly favor disordered networks [43,46,47], in our systems, the shape anisotropy still plays an important role, thus leading to a non-trivial assembly scenario.

The interplay between flexible bonds and non-spherical particle shapes can be rationalized via a thorough analysis of the different bond types emerging on increasing the bond flexibility. On increasing the patch size, two types of bonds appear in the systems: multiple bonds per patch and multiple bonds per particle pair. In the first bond type, a patch can form a bond with more than one patch; in the second, pairs of particles can be bonded to each other via more than one patch.

We show that when the combination of the patch size and patch position allows for the formation of two/three bonds per patch, parallel domains are promoted over non-parallel ones. When the patch size is increased even further, patches can—depending on their position along the edges—bond with up to four/five other patches; in this case, mixed bonds emerge (such as five- and six-star configurations) that completely destroy the ordering of the surface structures. The described crossover from the non-parallel to parallel domains and, further, to disordered structures is the more significant the more the patch topology is asymmetric, as off-center patches are more accessible to other off-center patches. While poly-bonds per patch allow for a high degree of bond flexibility, poly-bonds per particle pair diminish it: as soon as two particles bond together twice, their ability to wiggle around is strongly reduced, and particles prefer to bond in a parallel fashion; as poly-bonds per particle pair co-occur with poly-bonds per patch, they contribute to the emergence of parallel patterns in the tilings.

In summary, we perform a thorough investigation of the tiling mechanisms occurring in systems of patchy, non-spherical platelets on increasing their bond flexibility; our description in terms of single versus multiple bonds between pairs of particles and/or patches provides a road-map to possibly promote the self-assembly of target ordered phases at different length scales.

## Figures and Tables

**Figure 1 ijms-21-08621-f001:**
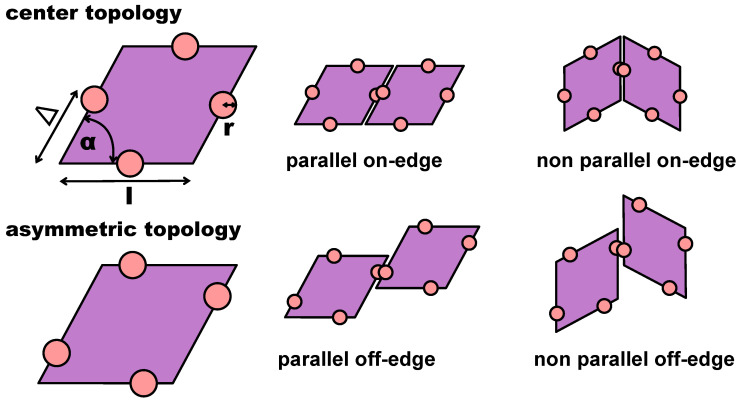
**Top row:** (left) sketch of a rhombic platelet with four patches placed in the center of each edge: the edge size *l* sets the unit length; the opening angle is fixed at α=60°; the patch size is defined by *r*; the patch position is determined by Δ, as labeled; (center) bonded pair in the parallel on-edge configuration; (right) bonded pair in the non-parallel on-edge configuration. **Bottom row:** (left) sketch of a rhombic platelet with four patches arranged in the chosen asymmetric topology with Δ<0.5; (center) parallel off-edge configuration; (right) non-parallel off-edge configuration.

**Figure 2 ijms-21-08621-f002:**
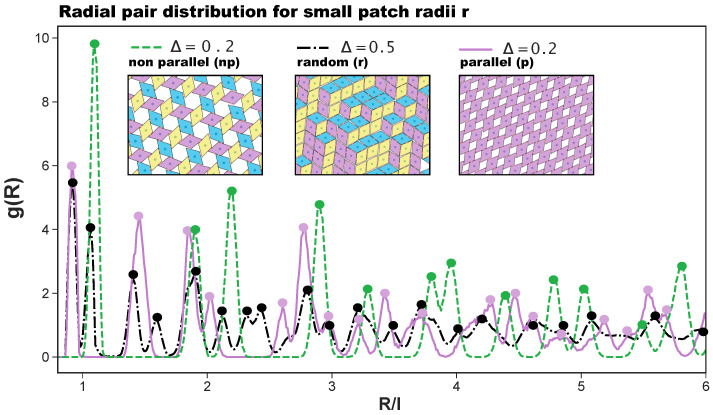
Radial distribution function g(R), where *R* is the distance between the centers of mass of the rhombi, for (i) the porous non-parallel (np) tiling (dashed green line), (ii) the porous parallel (p) tiling (continuous purple line), and (iii) the random (r) tiling (dashed and dotted black line). The first two tilings—as they emerge at Δ=0.2 and r=0.05 [27]—are sketched in the left and the right insets, respectively, as labeled; the r-tiling—as it emerges at Δ=0.5 and r=0.05 [27]—is sketched in the central inset, as labeled. The colors in the three snapshots highlight the particle orientations.

**Figure 3 ijms-21-08621-f003:**
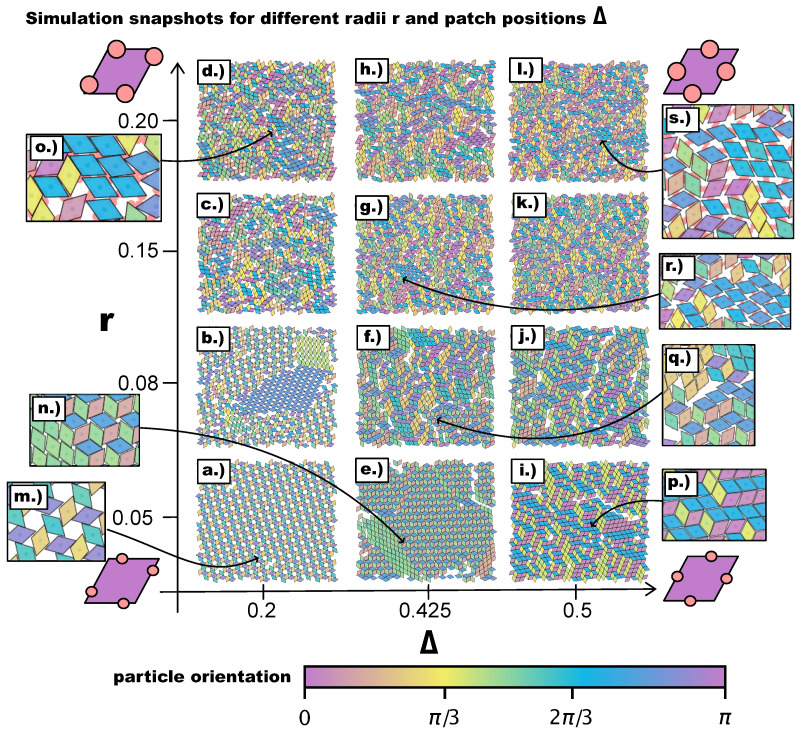
Simulation snapshots of the self-assembly products for different values of the patch position Δ=0.2,0.425,0.5 and the patch radius r=0.05,0.08,0.15,0.2 after ≈$10^{7}$ Monte Carlo sweeps. Particle colors represent the absolute particle orientation, which is obtained by arbitrarily defining a zero orientation at the beginning of the simulation: when the simulation is initialized—before the equilibration—all particles have the same orientation, which is then used as a reference. The reference orientation is represented in the particle sketches in purple, and the color scheme is chosen such that orientations that differ by π/3 are highlighted. (**a**–**d**) Self-assembled structures at Δ=0.2 and r=0.05,0.08,0.15,0.2. (**e**–**h**) Self-assembled structures at Δ=0.425 and r=0.05,0.08,0.15,0.2. (**i**–**l**) Self-assembled structures at Δ=0.5 and r=0.05,0.08,0.15,0.2. (**m**)–(**s**) Zooms of snapshots as highlighted by the arrows: (**m**) Zoom of snapshot (**a**). (**n**) Zoom of snapshot (**e**). (**o**) Zoom of snapshot (**d**). (**p**) Zoom of snapshot (**i**). (**q**) Zoom of snapshot (**f**). (**r**) Zoom of snapshot (**g**). (**s**) Zoom of snapshot (**l**).

**Figure 4 ijms-21-08621-f004:**
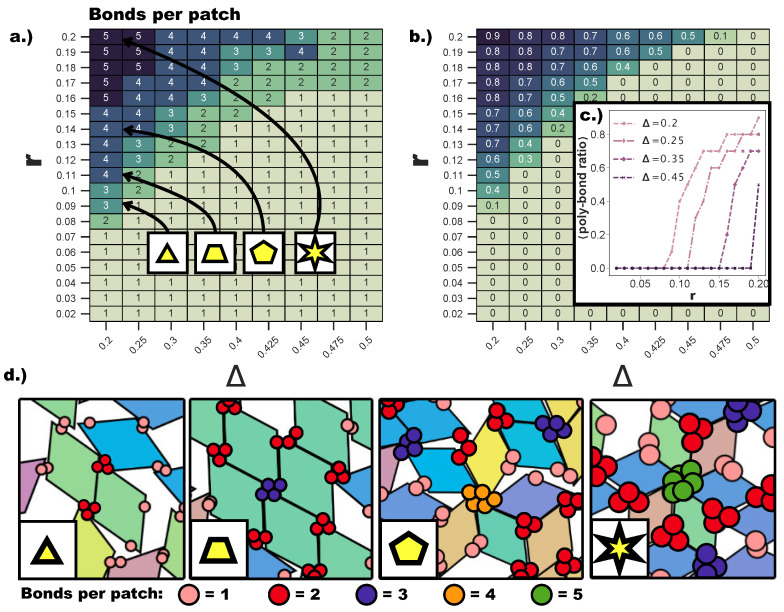
(**a**) Heat map of the maximum number of bonds per patch for Δ∈[0.2,0.5] and r∈[0.02,0.2]. The maximum number of bonds per patch is calculated (see Section 4) over all patches in 10 configurations taken after ≈$10^{7}$ Monte Carlo sweeps of each of the 16 parallel runs conducted for each value of (*r*,Δ). (**b**) Heat map of the average fraction of patches with more than one bond per patch—i.e., the average poly-bond per patch ratio—within one configuration for systems with Δ∈[0.2,0.5] and r∈[0.02,0.2]. The average is taken over all configurations used for calculating (**a**). (**c**) The average poly-bond per patch ratio as a function of *r* for Δ=0.2,0.25,0.35,0.45. (**d**) Snapshots to illustrate typical single- and poly-bonded configurations. The number of bonds per patch is indicated in different patch colors: one bond per patch (salmon), two bonds per patch (red), three bonds per patch (blue), four bonds per patch (orange), five bonds per patch (green). The labels (triangles, trapezoid, pentagon, star) indicate the (Δ,r)-values of the systems from which the snapshot was taken. Note that the number of patches per patch aggregate is by definition greater than the number of bonds per patch.

**Figure 5 ijms-21-08621-f005:**
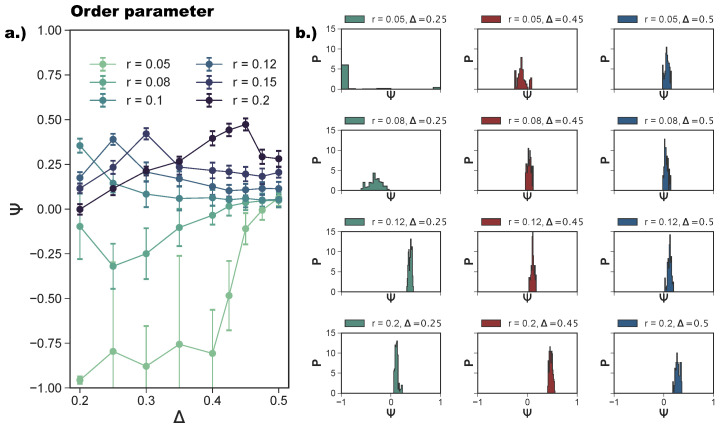
(**a**) The average order parameter Ψ (see Section 4) of the largest cluster in the system for different patch radii r=0.05,0.08,0.1,0.12,0.15,0.2 as a function of the patch position Δ; the average is performed over 16 parallel runs and taken over 10 simulation checkpoints each after ≈$10^{7}$ Monte Carlo sweeps. (**b**) Histograms for Ψ at Δ=0.25,0.45,0.5 and r=0.05,0.08,0.12,0.15 generated by 16 parallel runs and taken over 10 simulation checkpoints after ≈$10^{7}$ Monte Carlo sweeps for each system.

**Figure 6 ijms-21-08621-f006:**
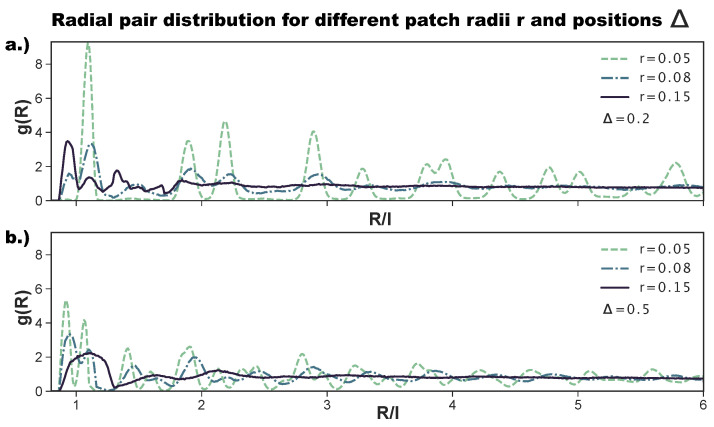
Radial distribution function g(R) for different patch radii *r* (as labeled) at two different Δ-values, where *R* is the center-to-center distance between two rhombi and *l* is the rhombi side length. (**a**) Δ=0.2, (**b**) Δ=0.5. Data are averaged over 16 simulation runs.

**Figure 7 ijms-21-08621-f007:**
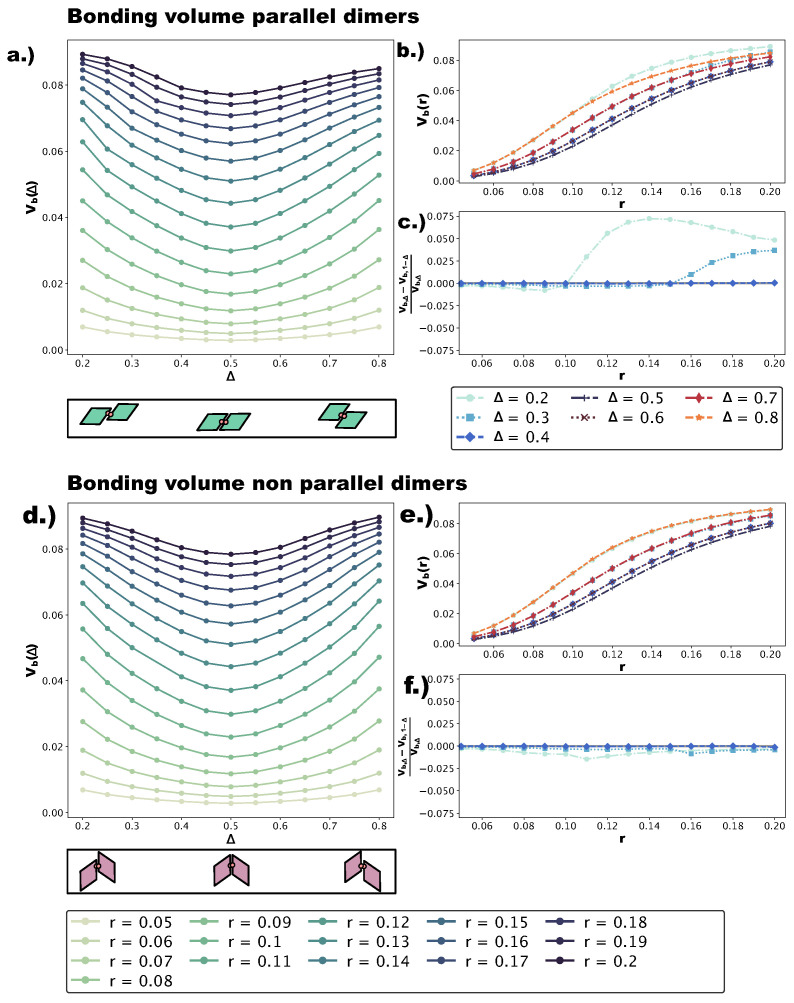
(**a**) Bonding volume $V_{b}$ for p-bonded dimers as a function of Δ∈[0.2,0.8] at several *r*-values ∈[0.05,0.2]. $V_{b}$ is calculated via a Monte Carlo simulation of two bonded particles: the dimer is not allowed to break over $2\cdot 10^{8}$ configurations (see Section 4). A sketch of the p-bonded dimer at Δ=0.2,0.5,0.8 is given below the plot. Note that the configuration at Δ=0.2 is qualitatively different from the configuration at Δ=0.8: at Δ=0.2, the rhombi are bonded such that the small angles are close to each other, while at Δ=0.8, the rhombi are bonded such that the big angles are close to each other. (**b**) $V_{b}$ for p-bonded dimers at different Δ∈[0.2,0.8] as a function of r∈[0.05,0.2]. (**c**) The asymmetry of $V_{b}$ for p-bonded dimers with respect to Δ=0.5, as a function of *r*, calculated as [$V_{b}\left(\Delta \right)-V_{b}\left(1-\Delta \right)]/V_{b}\left(\Delta \right)$. (**d**) $V_{b}$ for np-bonded dimers as a function of Δ∈[0.2,0.8] at several *r*-values ∈[0.05,0.2]. A sketch of the np-bonded configurations at Δ=0.2,0.5,0.8 is given below the plot. Note that the configurations Δ=0.2 and Δ=0.8 are just mirrors of each other with respect to Δ=0.5. (**e**) $V_{b}$ of np-bonded dimers at different Δ∈[0.2,0.8] as a function of r∈[0.05,0.2]. (**f**) The asymmetry of $V_{b}$ for np-bonded dimers with respect to Δ=0.5, as a function of *r*, calculated as [$V_{b}\left(\Delta \right)-V_{b}\left(1-\Delta \right)]/V_{b}\left(\Delta \right)$.

**Figure 8 ijms-21-08621-f008:**
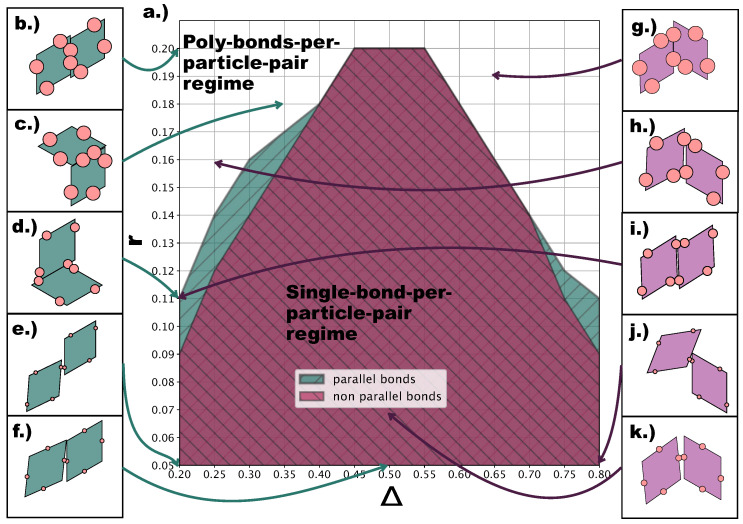
(**a**) Heat map indicating the presence of poly-bonds per particle pair at different (Δ, *r*)-values. A particle pair is called poly-bonded if it is bonded through more than one patch; otherwise, a particle pair is called single-bonded. To determine whether a particle pair has the ability to bond more than once, we run a Monte Carlo simulation of a particle pair initialized in a single-bonded state (see Section 4). We declare particle bonds as initially parallel (turquoise) if they are initialized with a parallel single bond and initially non-parallel (burgundy) if they are initialized with a non-parallel single bond. (**b**–**f**) Snapshots of configurations of initially parallel bonded particle pairs. (**g**–**k**) Snapshots of configurations of initially non-parallel bonded particle pairs.

**Figure 9 ijms-21-08621-f009:**
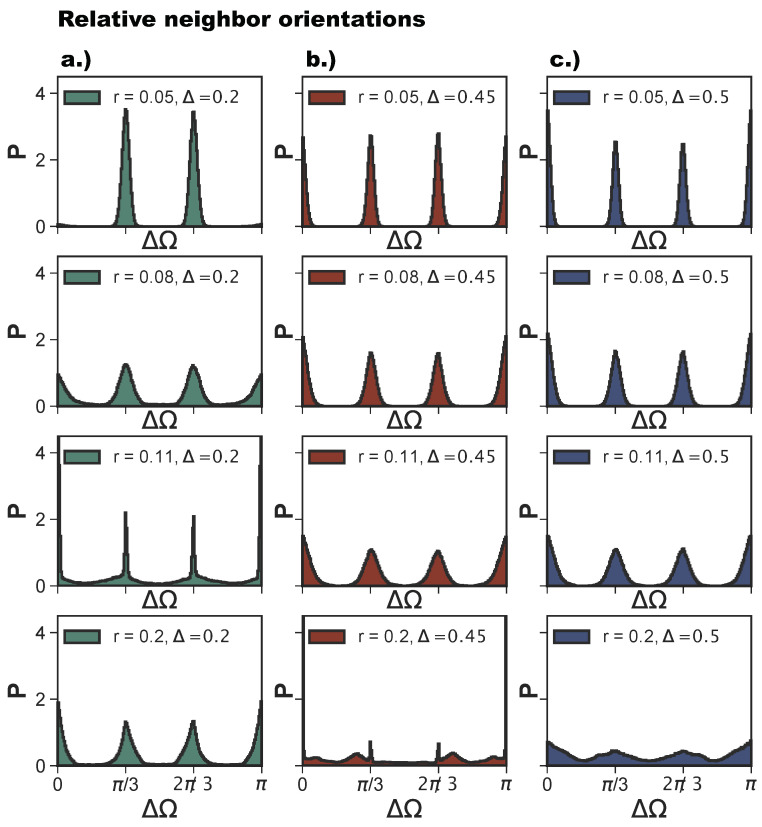
Histograms of relative orientations of bonded neighbors, ΔΩ, at selected radii *r* (rows) and patch positions Δ (columns): r=0.05,0.08,0.11 and Δ=0.2 (green, panel **a**), Δ=0.45, (red, panel **b**) and Δ=0.5 (blue, panel **c**). The histograms are calculated over 10 configurations and 16 parallel runs, after ≈$10^{7}$ Monte Carlo sweeps (see Section 4).

**Figure 10 ijms-21-08621-f010:**
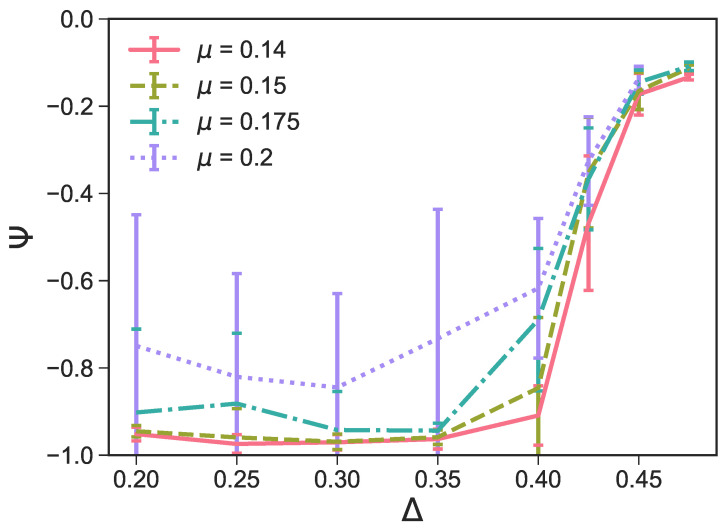
The average order parameter Ψ within the largest cluster as a function of the patch position Δ for r=0.05 and different chemical potentials μ∈[0.14,0.15,0.175,0.2]. The average is taken over the last 10 checkpoints of 16 parallel runs for each system, and the error bars denote the standard deviation of these samples. As the purpose of this graph is to measure the number of parallel bonds within non-parallel samples, configurations yielding highly parallel tilings—i.e., with Ψ>0.8—have been excluded from the evaluation.

**Table 1 ijms-21-08621-t001:** Particle parameters. For the definition of the geometric parameters, see the particle sketch reported at the top-left corner of Figure 1. All length parameters are in units of the edge length *l*. Note that throughout the text, we use patch radius and patch size interchangeably.

Variable	Symbol	Values
Angle	α	60°
Side length	l	1
Patch position	Δ	0.2–0.5
Patch radius/patch size	*r*	0.02–0.2
Interaction strength	ϵ	$5.2k_{B}T$

**Table 2 ijms-21-08621-t002:** Simulation parameters.

Variable	Symbol	Values
Initial chemical potential	$\mu^{eq}$	0.1
Chemical potential	$\mu^{*}$	0.14
Area	A	1000·sin(60)
Temperature	*T*	0.1

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
