# Peer review of "A Matter of Size and Placement: Varying the Patch Size of Anisotropic Patchy Colloids"

_ijms, 2020, doi:10.3390/ijms21228621_

Round 1

Reviewer 1 Report

The manuscript "A Matter of Size and Placement - Varying the Patch Size of Anisotropic Patchy Colloids" by Bianchi and co-workers describes the theoretical investigation of the effect of the patch size on the two-dimensional tiling of an anisotropic patchy colloids. The authors use a logically constructed study design based on a well-known set of methods while varying certain patch parameters. The obtained results will help explain and help predict the properties of self-assembly and construction of various supramolecular systems from a fundamental point of view.

The manuscript is well written. I recommend the authors to make a separate section "Conclusions", in which the important obtained results in the work should be written in a clear text form for a non-specialist in this field. This will increase interest in reading the manuscript, and also open up opportunities for interdisciplinary use of the results.

Author Response

We thank the referee for his/her appreciation of our work and for the wise suggestion he/she gave us. In the revised manuscript we now have a section “Discussion” where we describe the insights our theoretical/numerical work offers for the understanding of experimental results, and a section “Conclusions” where we summarize the major findings of our investigation. The latter part has been revised to make it more accessible to a broad audience. The key part of our summary now reads as:

We show that when the combination of patch size and patch position allows for the formation of two/three bonds per patch, parallel domains are promoted over non parallel ones. When the patch size is increased even further, patches can – depending on their position along the edges – bond with up to four/five other patches; in this case, mixed bonds emerge (such as 5- and 6-star configurations) that completely destroy the ordering of the surface structures. The described crossover from non parallel to parallel domains and, further, to disordered structures is the more significant the more the patch topology is asymmetric, as off-center patches are more accessible to other off-center patches. While poly-bonds per patch allow for a high degree of bond flexibility, poly-bonds per particle pair diminish it: as soon as two particles bond together twice, their ability to wiggle around is strongly reduced and particles prefer to bond in a parallel fashion; as poly-bonds per particle pair co-occur with poly-bonds per patch, they contribute to the emergence of parallel patterns in the tilings.

Reviewer 2 Report

Comments included in the attached document.

Author Response

We thank the referee for his/her appreciation of our work and for the detailed feedback he/she gave us. We carefully considered all the received comments and modified the manuscript accordingly. We attach our point-to-point answers, together with the description of the changes we have carried out throughout the manuscript.
